# Diabetic Kidney Disease: Goals for Management, Prevention, and Awareness

## Callie W. Greco and Julianne M. Hall *

Frank H. Netter School of Medicine, Quinnipiac University, Hamden, CT 06518, USA; callie.greco@quinnipiac.edu
* Correspondence: julianne.hall@quinnipiac.edu

**Definition:** Diabetic kidney disease (DKD), which is diagnosed on the basis of reduced glomerular filtration rate (GFR), increased albuminuria, or both, is the leading cause of chronic kidney disease (CKD) and end-stage renal disease (ESRD) worldwide. Future projections anticipate a significant increase in diabetes cases, with close to 700 million diabetes patients internationally by the year 2045. Amidst ongoing research into novel biomarkers and therapeutic agents for DKD, the current clinical preventative strategy for DKD involves (1) intensive glycemic control, (2) treatment of associated co-morbidities (hypertension and hyperlipidemia), and (3) instruction on lifestyle modifications, including smoking cessation, exercise, and dietary habits. In addition to these three categories, patient education on renal injury, a fourth category, is equally important and necessary in the collaborative effort to reduce global rates of DKD. In this entry, authors highlight and discuss these four core categories for DKD prevention.

**Keywords:** diabetes; chronic kidney disease; diabetic kidney disease; prevention; glycemic index; patient education; lifestyle modification

## 1. Introduction

Chronic kidney disease (CKD) is defined as the long term, progressive decline in kidney function due to accumulated renal damage. As renal injury accrues, CKD can eventually transition to its final stage, a devastating and costly complication, referred to as end-stage renal disease (ESRD). At ESRD, one of two permanent measures must be implemented to replace the failing kidneys. Patients will either need a renal transplant or an initiation of life-long hemodialysis therapy.

Future projections anticipate a significant increase in diabetes cases, with close to 700 million diabetes patients internationally by the year 2045 [1]. According to the National Institute of Diabetes and Digestive and Kidney Diseases (NIDDK), there are several risk factors which can contribute to CKD development. These include drug-induced or infection-induced kidney injury, systemic autoimmune diseases, such as lupus and Goodpasture's syndrome, or genetic conditions, like polycystic kidney disease [2]. However, the two most common causes for renal injury leading to CKD are hypertension and diabetes, with the latter being the leading cause of CKD [2,3]. Globally, diabetic nephropathy (DN), commonly referred to as diabetic kidney disease (DKD), contributes to over two-thirds of all CKD cases [4].

Diabetic kidney disease (DKD) is defined by both physiological and structural pathologic alterations. Three classic maladaptive renal changes are seen on histologic examination: nodular glomerulosclerosis, mesangial cell expansion, and fibrosis of the glomerular and tubular basement membranes [5,6]. The progressive parenchymal injury interferes with the vital metabolic and homeostatic functions of the kidney. As a result, the objective evidence used for DKD diagnosis involves a gradual decrease in the estimated glomerular filtration rate (eGFR) and progressive micro- to macro-albuminuria [7]. GFR and urinary albumin levels are categorized and correlated with the sequential CKD stages (1 through 4)

and ESRD (stage 5). Among those with type 1 diabetes, up to 40% of patients are diagnosed with DKD within 15–20 years of diabetes onset [8]. For type 2 diabetes patients, the UK Prospective Diabetes Study (UKPDS) reported that more than 50% of patients develop DKD after an average of 15 years post-diabetes diagnosis [9]. As many as 20% of newly diagnosed type 2 diabetes patients already have signs of renal disease at the time of their diabetes diagnosis [10]. There also exists a prolonged asymptomatic phase of DKD, where renal injury is occurring but at a level undetectable by serum and urinary screening tests [11].

DKD is unfortunately a very prevalent consequence of long-term inadequate disease control in both type 1 and type 2 diabetes populations. The 2017 National Diabetes Statistics Report released by the Center for Disease Control (CDC), estimated that in 2015 diabetes affected approximately 30.3 million people of all ages (9.4% of the U.S. population). Among all individuals with diabetes, type 2 diabetes accounted for about 95% of the cases [12]. There is a similar concerning upward trend in diabetes worldwide, especially with type 2 diabetes in low- and middle-income countries largely due to the ongoing obesity pandemic [12,13]. The International Diabetes Federation (IDF) recently published the 2019 results on current global diabetes rates, including future disease projections. The report identified that 436 million adults, ages 20–79, are currently living with diabetes. Furthermore, the IDF report estimated that by the year 2045, there will be near 700 million people worldwide with diabetes [13].

Even more troubling is the simultaneous rise in the diabetes-associated mortality rate, which in 2019, accounted for 4.2 million deaths globally [13]. Diabetes is known to significantly reduce total life expectancy compared to non-diabetic populations. In fact, current statistics suggests a 6-year decrease in total life expectancy in those with a diabetes diagnosis [14]. Within the diabetes population, cardiovascular (CV) disease (coronary heart disease, stroke, heart failure) remains a prevalent cause of death. Although the majority of diabetes deaths are the result of CVD, recent reports have highlighted the steady incline in DKD-attributed mortality. A risk analysis study published by the American Diabetes Association (ADA) found an increase in all-cause mortality rates in diabetes cohorts compared to non-diabetes cohorts, even after adjusting for age, BMI, systolic blood pressure, and total and HDL cholesterol [15]. Specifically, there was a strong positive correlation between chronic diabetes and death from associated complications, including renal nephropathy and fatty liver disease.

In an effort to reduce the global burden of DKD, as well as DKD-associated mortality, it is critically important to address the risk factors for DKD in diabetes populations. The primary modifiable risk factors for DKD include uncontrolled glycemic index, hypertension and hyperlipidemia co-morbidities, and lifestyle factors (smoking, poor nutrition, and minimal exercise) [5,16,17]. Since the incidence of diabetes is expected to significantly rise over the next couple of decades, it is equally necessary to inform and educate at-risk diabetes patients on the potential end-organ complications resulting from inadequate diabetes management. This entry will review the current preventative approach for DKD, including strict glycemic control, treatment of hypertension and hyperlipidemia, promotion of patient-driven lifestyle modifications, and incorporation of point-of care patient education.

## 2. Glycemic Index

Management of serum glucose in patients with diabetes remains a priority in the effort to minimize microvascular and parenchymal kidney damage. The direct pathophysiological link between dysregulated glucose and kidney disease, as well as disease biomarkers, has been established. Specifically, there is general consensus on the harmful role of advanced glycosylation end product (AGEs), which are produced from the non-enzymatic covalent linkage of excess glucose with serum protein and lipids [18–21]. Chronic hyperglycemia in diabetes patients increases the rate of AGEs formation, and thus raises the likelihood for AGEs-associated end organ damage [19].

Accumulation of AGEs along the vasculature can perturb normal endothelial cell function by interacting with advanced glycosylation end product receptors (RAGEs) and triggering generation of reactive oxygen species (ROS) [20]. Chronic microvascular injury via AGEs in the kidney can eventually progress to macrovascular damage. This large vessel degradation leads to a potentially rapid and spiraling pathway to kidney failure. Specifically, macrovascular damage results in diminished afferent renal perfusion. The ensuing downstream hypoxia in glomerular and tubular structures contributes to the classic histological findings of DKD, including glomerulosclerosis and tubular necrosis [5,19]. As parenchymal damage accrues, there is a hemodynamic shift in renal blood flow, leading to the overwhelming hyperperfusion of any surviving nephrons [22]. The evolving vascular damage by AGEs, as well as the compensatory parenchymal damage, continues to accumulate and correlates to CKD stage progression.

Intense glucose regulation in diabetes patients is a cornerstone in the effort to delay vascular damage in the kidney. Hemoglobin A1C (HbA1C) is a serum test that assesses the average amount of serum glucose over the past three months by measuring the percentage of non-enzymatic glycated hemoglobin. Since red blood cells have a lifespan of 2–3 months, the HbA1C is used to routinely monitor diabetes patients and modify their treatment plans in order to reach glycemic goals. According to the 2020 Standards of Medical Care in Diabetes, distributed by the American Diabetes Association (ADA), the strict management of HbA1C to near-normoglycemia levels (<7%) was correlated with a delay in the onset of albuminuria and decreased eGFR in both type 1 and type 2 diabetes patients [7]. Likewise, the Diabetes Control and Complication Trial (DCCT) compared intense (goal HbA1C < 7%) glycemic control with conventional (mean HbA1C 9.1%) management, and discovered a 39% decrease in microalbuminuria progression among type 1 diabetes patients [23].

Another study, the Action in Diabetes and Vascular Disease: Preterax and Diamicron MR Controlled Evaluation Trial [24], enrolled and randomized over 11,000 patients with type 2 diabetes into intense (goal HbA1C 6.5% or less) and standard (goal HbA1C 7.3%) glycemic control cohorts. After five years, the study reported overall improved kidney outcomes in the intense glycemic control group, with decreases in new onset micro- and macro-albuminuria and an astonishing 65% decrease in ESRD.

There are many pharmacological agents used in practice today to establish intense glycemic control in type 1 and type 2 diabetes patients [7]. Insulin therapy is required in all type 1 and some advanced type 2 diabetes patients. The noninsulin options continue to expand in variety and each target a slightly different mechanisms in the glucose homeostasis pathway. Insulin sensitizers (biguanides) such as metformin, achieve glycemic control by inhibiting hepatic gluconeogenesis and opposing the actions of glucagon. Incretin agonists (GLP-1 receptor agonists) bind to receptors on pancreatic β-cells where they facilitate glucose-dependent insulin release. Renal sodium-glucose cotransporter-2 (SGLT-2) inhibitors lower blood sugar by preventing reabsorption of filtered glucose, resulting in excess sugar loss in the urine. As with many therapeutic interventions, the risk of adverse events needs to be considered before and during patient treatment. The primary and most dangerous complication from intense glycemic control is severe or fatal hypoglycemia [25]. The Action to Control Cardiovascular Risk in Diabetes (ACCORD) [26,27] trial reported increased risk of mortality from severe hypoglycemia in type 2 diabetes patients following intense glycemic control (median HbA1c 6.4%).

In summary, intense glycemic control appears to improve renal disease outcomes and should be implemented as early as possible in the treatment plans for type 1 and type 2 diabetes patients. Potentially fatal adverse events, including severe hypoglycemia, are of concern and must be closely monitored in patients following intense glycemic control regimens.

## 3. Patient Co-Morbidities

### 3.1. Hypertension

High blood pressure is linked with its own complex multi-system pathophysiology which can chronically exacerbate many diseases. Adaptations to the long-term and/or severe high-pressure flow involve vascular wall thickening, including hyaline and hyperplastic arteriosclerosis. The prolonged luminal narrowing within the vasculature can eventually cause hypoperfusion and hypoxia-induced end organ damage, including various nephropathies, retinopathies, cardiopathies, and damage to other tissues and organs [28]. In the kidneys, long-term hypoperfusion and hypoxia contributes to parenchymal injury, metabolic dysfunction, and further accelerates hyperperfusion injury in any remaining viable nephrons [29].

In diabetic patients, hypertension adds another dimension in the management and prevention of DKD. Compared to the general population, diabetes patients are twice as likely to be hypertensive [30]. Several placebo-controlled trials have analyzed the risks and benefits of intense blood pressure management in diabetes patients with diagnosed hypertension. Recent studies have shown that intense blood pressure therapy is associated with delayed albuminuria, as well as decreased rates of serious extra-renal cardiovascular events [31–33]. The UK Prospective Diabetes Study (UKPDS) demonstrated that reduction in systolic blood pressure (154 mmHg to 144 mmHg) in type 2 diabetes patients with newly diagnosed hypertension was associated with a 29% decrease in microalbuminuria onset [34].

Current first-line therapy for diabetes patients with hypertension include agents that block the renin-angiotensin-aldosterone system (RAAS): angiotensin converting enzyme inhibitors (ACEi) and angiotensin II receptor blockers (ARBs) [7,35]. Other frequently used options are thiazide diuretics and calcium channel blockers. Dosage and combination patterns should take into account the patient's CKD status and diabetes status.

Other emerging therapies for hypertensive diabetics include SGLT-2 inhibitors, such as empagliflozin and dapagliflozin. As mentioned above, these agents are in use to help with diabetic glycemic control, however, there also appear to be a number of cardiovascular benefits. All current studies have shown significant reductions in systolic and diastolic blood pressure, with a similar efficacy observed in patients with compromised renal function [36]. Furthermore, the EMPA-REG OUTCOME trial demonstrated a 38% relative risk reduction in death from cardiovascular causes in patients with established cardiovascular disease who were administered empagliflozin [37].

There is an ongoing debate regarding the intensity level for blood pressure management in patients with DKD. The therapeutic target for systolic and diastolic pressures in general practice is <140/90 mmHg, but the ideal systolic and diastolic pressure levels are still undetermined. The observation that there exists an increased risk of death or ESRD of 6.7% for every 10 mm Hg increase in baseline systolic blood pressure prompted the consideration of whether high-intensity pharmacological antihypertensive therapy (concommittent use of an ACEi and ARB) could be beneficial. However, while standard antihypertensive therapy (ACEi or ARB) achieved a decrease in proteinuria and other primary renal endpoints, high-intensity pharmacological blood pressure therapy was associated with increased rates of hyperkalemia and acute kidney injury (AKI) [3,8,32]. Therefore, the general consensus is to avoid combination therapy with an ACEi and an ARB, due to high rates of adverse side effects.

It is also not clear whether therapy targeted at achieving lower blood pressure goals than the standard goal of <140/90 mmHg is indeed beneficial. Data from the previously mentioned ACCORD trial found that 3% of diabetes patients with no evidence of CKD at baseline were diagnosed with CKD after three years of a systolic blood pressure treatment target of less than 120 mm Hg, compared to 1% of patients with standard antihypertension therapy goals of <140/90 mmHg [38]. Yet, the use of antihypertension therapy has been shown to reduce cardiovascular events, such as stroke risk and myocardial infarction [31,34]. In summary, the data suggests that identifying an optimal blood pressure must consider

the patient's personal cardiovascular status and CKD stage, in order to minimize any deleterious risks.

### 3.2. Hyperlipidemia

In DKD, the impaired filtration function from chronic renal injury contributes to a more atherogenic serum lipid profile, including hypertriglyceridemia and low HDL [39]. Likewise, sustained microvascular changes, as discussed previously, create an opportune environment for accelerated atherosclerotic deposition and accumulation in the endothelium. As a result, hyperlipidemia treatment is another consideration in diabetes patients.

The major benefit of lipid treatment in diabetes patients is the reduced rate of adverse cardiovascular (CV) events [40,41]. As previously discussed, the risk for CVD and CVD-related death is significantly high in the DKD population, and thus treatment for hyperlipidemia is usually strongly indicated. The clinical guidelines released by the American Diabetes Association suggest a target LDL of <100 mg/dL for diabetes patients or LDL < 70 mg/dL for diabetes patients with established cardiovascular disease [42].

Statins are the current pharmacological standard for dyslipidemia treatment [43]. The Collaborative Atorvastatin Diabetes Study (CARDS) showed that use of atorvastatin significantly decreased cardiovascular events in diabetes patients with at least one coronary disease risk factor [39,44]. Similarly, a systematic review found that lowering LDL cholesterol decreased total atherosclerotic events in non-dialysis CKD and renal transplant patients [39].

Despite substantial evidence for statin benefits, the majority of DKD patients continue to suffer from severe cardiovascular disease. There is a need for new therapeutic approaches for dyslipidemia in DKD populations, beyond the primary treatment modalities of statins, statin-ezetimibe combinations, and elimination of high-fat diets [42]. There are new non-statin drugs that demonstrated extreme promise in clinical trials and have now been routinely incorporated into treatment plans for patients who do not achieve sufficient lipid-lowering effects from tolerable doses of statins. The low-density lipoprotein (LDL) receptor (LDLR), located on the hepatocytes, binds and removes LDL cholesterol (LDL-C) from circulation. The proprotein convertase subtilisin/kexin type 9 (PCSK9) inhibitor blocks the recycling of the LDLR in the liver. Several clinical trials have reported on the efficacy on PCSK9 inhibitors in dyslipidemia treatment for patients with and without diabetes [45–47]. In 2015, the U.S. Food and Drug Administration (FDA) approved two PCSK9 inhibitors, alirocumab and evolocumab, which are derived from human monoclonal antibodies. Despite the potent lipid lowering capacity, PCSK9 inhibitor therapy requires subcutaneous injections every two or four weeks and can be accompanied by adverse events, such as myalgias and neurocognitive effects [48].

Another non-statin alternative for dyslipidemia receiving considerable attention is bempedoic acid (BA). Bempedoic acid is an inhibitor of ATP citrate lyase, an enzyme upstream in the cholesterol synthesis pathways [49]. Early clinical trial results showed that bempedoic acid, alone or in conjunction with statins and/or ezetimibe, reduced LDL cholesterol by a range of 17–64% [49]. Another trial tested a combination of bempedoic acid and ezetimibe (BA + EZE) in type 2 diabetes patients with cardiovascular risk [50]. The researchers found a significant decrease in LDL cholesterol in patients receiving the BA + EZE combination, compared to those that received ezetimibe alone or a placebo. In February 2020, the FDA approved a once daily oral dose of bempedoic acid (Nexletor) as an adjunct to maximally tolerated statins in refractory hyperlipidemia cases. Unlike statins, which frequently cause mild to severe myalgias, bempedoic acid does not appear to have any significant adverse side effects [49,50]. This is attributed to the fact that bempedoic acid is a pro-drug, which is not metabolized to an active form in skeletal muscle, whereas statins are known to exert numerous deleterious effects in myocytes via their ability to disrupt calcium signaling and cell membrane integrity.

In summary, treatment for hyperlipidemia in DKD patients is recommended due to its mitigating effects on cardiovascular disease.

### 3.3. Congestive Heart Failure

Guideline-driven medical treatment of symptomatic heart failure includes pharmacological management with several agents that also display efficacy in achieving glycemic control and preventing CKD progression in type 2 diabetics. In 3 independent clinical trials, patients taking SGLT-2 inhibitors demonstrated a significant (20% to 25%) relative reduction in hospitalization for heart failure or cardiovascular death when compared with those administered placebo. This benefit was consistent among patients with or without type 2 diabetes and in those experiencing CKD. Patients also reported a marked improvement in health-related quality of life surveys [51].

Overactivation of the mineralocorticoid receptor is a common finding in both cardiovascular and renal disease. Finerenone is a selective nonsteroidal mineralocorticoid receptor antagonist that is now also used as a standard therapy for symptomatic congestive heart failure. Notably, there is increasing evidence that finerenone can prevent heart failure exacerbations and associated hospitalization in type 2 diabetic patients with various stages and severities of CKD. Furthermore, a reduction in other adverse cardiovascular outcomes including death from cardiovascular causes, myocardial infarction, or stroke was observed in patients taking finerenone [52–54].

### 3.4. Metabolic Syndrome

In addition to localization in pancreatic islets, incretin receptors have a wide range of tissue distribution, including the vasculature, cardiac myocytes, adipose tissue, gastrointestinal tract, hepatocytes, and central and peripheral nervous systems. Thus, in addition to augmentation of insulin secretion, GLP-1 agonists appear to have pleiotropic and beneficial effects on a wide range of metabolic functions. Their efficacy in metabolic syndrome, a common co-morbidity in type 2 diabetes may be attributed to many mechanisms. GLP-1 agonists are effective in altering blood lipid metabolism in a manner that lowers circulating LDL-C and reduces atherosclerosis and associated cardiovascular disease. Furthermore, GLP-1 agonists have potent anti-inflammatory effects, which may explain their efficacy in cardiovascular fitness, prevention of stroke, alleviation of chronic pain, and ability to reduce circulating levels of liver enzymes and liver steatosis.

Several possible models have been proposed about mechanisms by which these GLP-1 agonists can display anti-obesogenic effects. In adipose GLP-1 receptor activation increases beta oxidation and can aid in the conversion of white adipose tissue to brown adipose tissue, thus promoting energy consumption. GLP-1 agonists also are effective in appetite suppression, as they delay gastric emptying, and are thought to have central actions in the arcuate nucleus of the hypothalamus in the stimulation of satiety [55].

## 4. Lifestyle Modifications

### 4.1. Diet

#### 4.1.1. Protein

Dietary protein recommendations for DKD patients are a controversial and heavily debated topic. Protein intake is important for systemic metabolic needs, but at high intake levels, can exacerbate albuminuria and renal injury in DKD patients [56]. Determining the optimal protein intake for DKD patients is complicated by the inherent challenges that often accompany nutritional studies, including patient adherence to meal plans, small sample sizes, and short timelines. As a result, few conclusive reports exist, and most clinical trials have produced contradictory results.

A 2008 meta-analysis on low-protein diets in DKD assessed thirteen randomized control trials. Compared to non-diabetic patients, the low protein diet (LDP) was associated with an overall slower rate of GFR decline in diabetes patients over time [57]. A more recent large-scale cohort study in Korea enrolled 1572 non-dialysis CKD patients and assessed dietary protein intake (DPI) and rate of CKD progression over four years [58]. The study also accounted for protein wasting energy (PWE) factors, such as muscle wasting,

malnutrition, and inflammation, which typically increase with CKD progression. Rather than DPI, the researchers reported PWE as closely associated with CKD progression.

Despite inconclusive reports, there are several clinical guidelines available which provide recommendations for dietary protein intake. The most recent 2020 Standard of Medical Care in Diabetes by the American Diabetes Association suggests that non-dialysis patients should consume 0.8 mg/kg body weight/day of protein [7]. The upper limit, 1.3 mg/kg body weight/day of protein, is discouraged due to increased rates of albuminuria and CVD-associated mortality [7].

### 4.1.2. Sodium

Dietary sodium reduction is broadly supported across diabetes and nephrology societies for use in DKD and CKD patients. Decreasing salt intake reduces systemic blood pressure and thus, minimizes hypertensive stress and injury in the kidneys [56]. The current recommendation for a low salt diet is less than 2300 mg/day salt intake [7]. Practical suggestions for patients include reducing salt use, opting for sodium-free food options, and less consumption of highly processed and prepackaged foods.

### 4.1.3. Fatty Acids

There is also evidence comparing DKD progression and the intake of saturated fats (SFA) verses polyunsaturated fats (PUFA). The Diabetes and Nutrition Clinical Trial (DCNT) in Spain correlated regression of nephropathy (decreased albuminuria) in diabetes patients with high PUFA and low SFA intake [59,60]. Likewise, the study found nephropathy progression (elevated albuminuria) in diabetes patients on a high SFA and low PUFA diet [59,60]. Although definitive evidence is lacking, the general suggestion is to replace food rich in trans-fatty and saturated-fatty acids with more omega-3 and omega-9 options, including seafood (salmon, mackerel), nuts (walnuts, almonds, cashews), and oils (olive, almond, and avocado) [42,61].

### 4.1.4. Dietary Patterns

The traditional Western diet is filled with unhealthy sugars, sodium, trans-fats, and carbohydrates [62]. As a consequence, consumption of the Western diet is associated with increased rates of obesity and chronic diseases. Although specific daily recommendations exist for each dietary component, patients can often struggle to fully adhere to each guideline. Instead of focusing on individual minerals and nutrients, some studies encourage a more comprehensive dietary pattern approach [42,43]. The Mediterranean diet and Dietary Approaches to Stop Hypertension (DASH) diet are two dietary patterns which replace the Western diet habits with a more nutritional alternative based on fresh vegetables, fruits, unrefined carbohydrates, seeds, etc. [43]. Focusing on overall patterns of consumption would minimize confusion on food selection and instead, support active patient participation.

### *4.2. Exercise*

Exercise level is almost always included in patient care plans, including DKD treatment, due to its whole-body, multi-system benefits. Improved weight management, increased aerobic and cardiovascular capacity, and decreased inflammation and muscle atrophy, are all advantageous benefits from exercise, which may help to delay chronic disease progression among diabetes patients [63,64].

Few studies provide conclusive data on exercise therapy in DKD or CKD. Recommendations for specific types of exercise training programs in patient with CKD remain undetermined. A 2014 study reported improved muscular strength, walking capacity, and mobility in non-dialysis CKD patients after a twelve-week exercise program with varying exercise type, duration, and intensity [65]. Another randomized control study [66] enrolled non-dialysis CKD patients in stages 3–4. The experimental group following a home-based aerobic and resistance training program, including a daily 30 min walk and use of a handgrip strengthening device. The results demonstrated an increase in handgrip

and knee extension strength in the experimental group compared to the control. Although more research is needed, incorporation of an exercise routine in DKD and CKD treatment plans shows promising benefits.

### 4.3. Smoking Cessation

Several studies have identified tobacco use as an independent risk factor for renal disease. Toxic chemical exposure and chronic vasoconstriction by nicotine triggers vascular and parenchymal damage, including kidney fibrosis [33,67]. In diabetes patients, who are already at risk for renal injury, the additional damage from smoking can greatly increase their rate of progression to renal failure. Smoking can also significantly increase the risk and severity of cardiovascular events in diabetes patients [68]. Therefore, smoking cessation should be addressed as part of the preventative approach to DKD and associated co-morbidities.

### 4.4. Summary

In summary, individualized dietary and exercise modifications, as well as smoking cessation are all non-invasive, non-pharmacological therapies and should be considered an important factor in DKD prevention and management.

## 5. Patient Education

There is another aspect of diabetes clinical care that must be included and emphasized as part of the current DKD preventative strategy. The 2018 United States Renal Data System (USRDS) report revealed alarmingly low levels of disease awareness in patients with CKD. In patient populations with CKD and the risk factors (diabetes and hypertension), only 15% had knowledge about their risk of renal disease [69]. Furthermore, in a USRDS cohort from 2013–2016, only 10% of patients over 60 years old were aware of their CKD status [58]. These data unfortunately suggest that a substantial amount of DKD patients are going without basic education on their diagnosis, which can translate to inadequate intervention and poor therapy adherence. As a consequence, many DKD patients may needlessly suffer the devastating and debilitating process of permanent hemodialysis treatment. From a public health standpoint, a rise in ESRD cases requiring dialysis therapy will further contribute to escalating healthcare costs. In 2016, CKD and ESRD grossed $114 billion in Medicare spending, up an additional $16 billion from one year prior in 2015 [58].

These statistics can be interpreted from the provider and patient perspective. There are screening protocols in place for renal disease, especially in at-risk diabetes populations. Currently, the ADA recommends that all type 1 diabetes patients with a diagnosis $\geq 5$ years and all type 2 diabetes patients undergo annual urinary albumin and eGFR assessments [7]. Following these guidelines is crucial for identifying all patients at-risk for DKD and promptly intervening before additional renal injury occurs.

There is an even greater need to educate newly diagnosed diabetes patients on the long-term sequelae of their disease. A systematic review on self-management in patients with chronic illnesses, including diabetes, echoed the importance of patient education [70]. Two primary conclusions were reported: (1) Patient outcomes improved with incorporation of enhanced patient-oriented interventions, including education with guidebooks and care plans, and (2) Information must be provided at the time of diagnosis to promote self-care management and improve long-term health outcomes. The key is to target diabetes patients before DKD develops. Empowering patients with self-driven preventative strategies could have many potential advantages, including improved disease awareness, treatment adherence and ultimately, could contribute towards a reduction in global DKD rates. Online and in-office educational handouts and brochures are a simple and excellent option for clinical providers to initiate this discussion [70]. A multidisciplinary team-based approach, with participation by nutritional, occupational, and physical therapy experts, further strengthens the patient support network for more optimal outcomes.

In summary, patient-centered education on the importance of proper disease management could greatly improve general awareness of DKD and promote proactive intervention in diabetes patients.

## 6. Conclusions and Prospects

Diabetic kidney disease is a very common outcome of long-term, uncontrolled diabetes in both type 1 and type 2 diabetes populations. Although DKD has a gradual onset, there are several risk factors and associated diseases which can exacerbate renal injury and DKD progression. A thorough preventive approach for DKD includes glycemic control, treatment of hypertension and hyperlipidemia co-morbidities, and lifestyle modifications (diet, exercise, and smoking cessation) [5,16,17]. Patient education on DKD in the clinical setting is equally important and highly encouraged [70]. Since diabetes patients are initially asymptomatic, early intervention through patient education is an essential step in the collective goal to reduce the future DKD and ESRD burden.

**Author Contributions:** Conceptualization, C.W.G. and J.M.H.; methodology, C.W.G.; investigation, C.W.G.; writing—original draft preparation, C.W.G. and J.M.H.; writing—review and editing, J.M.H.; supervision, J.M.H. All authors have read and agreed to the published version of the manuscript.

**Funding:** This research received no external funding.

**Institutional Review Board Statement:** Not applicable.

**Informed Consent Statement:** Not applicable.

**Data Availability Statement:** Not applicable.

**Acknowledgments:** We thank the Scholarly Reflection and Concentration Course administrators at the Frank H. Netter MD School of Medicine for their continued support of medical student scholarship.

**Conflicts of Interest:** The authors declare no conflict of interest.

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
