# Peer review of "Diabetic Kidney Disease: Goals for Management, Prevention, and Awareness"

_encyclopedia, doi:10.3390/encyclopedia3030083_

Round 1
Reviewer 1 Report
An excellent review and very comprehensive. One of the better reviews I have read of diabetic chronic kidney disease
Author Response
We thank the Reviewer for their time, consideration, and overwhelmingly positive feedback on our manuscript!!
Reviewer 2 Report
SUMMARY
The authors provide a clear review of the pathophysiology and clinical relevance and scope of diabetic kidney disease (DKD). DKD is common amongst diabetics and may be attributable to poor glycemic control, co-morbid conditions, poor lifestyle behaviors, and lack of education. This review details several preventative measures (i.e., strict glycemic and blood pressure control, lipid management, and lifestyle modifications) that mitigate the risk of developing DKD. The authors also address patient education as the cornerstone of optimal patient care and outcomes. Although the information provided does not significantly advance the field pe se, the authors do a nice job of assimilating the major information related to DKD as it pertains to management, prevention, and awareness. The information is concise and presented in a manner in which non-expert readers would understand.
MAJOR STRENGTHS
The review is well-written and organized. It fulfills the aim of the Encyclopedia journal, which is to cumulate a comprehensive record of scientific information and evidence that could be interpreted by researchers and the public. The information presented throughout the review is considered established knowledge. The review is free of major grammatical errors and is easily readable. To the authors’ advantage, evidence from previous studies were included and utilized to leverage the information presented throughout the review.
SPECIFIC AREAS OF IMPROVEMENT
MAJOR POINTS
To make the review more relevant to current guideline alterations, the authors should consider including information about some of the newer therapeutic regimen for diabetes and congestive heart failure such as SGLT2 inhibitors, as these are co-morbid conditions that also cause DKD.
MINOR POINTS
On page 4, line 168 and elsewhere in the same paragraph, “angiotensin II receptor (ARB) inhibitors” should be changed to “angiotensin II receptor blockers (ARBs).” The term “ARB inhibitors” is a double negative and inaccurate.
On page 4, line 177, the authors should define what “high-intensity pharmacological blood pressure therapy” is. Also, a brief discussion about the mechanism for how it contributes to DKD should be provided.
On page 4, line 182, “…the use of hypertension therapy has shown…” should read “…the use of antihypertensive therapy has been shown…” The term “antihypertensive therapy” should also replace “hypertension therapy” in the preceding sentence.
On page 5, lines 214-217, the following two sentences should be reversed for clarity and context and read: “The low-density lipoprotein (LDL) receptor (LDLR), located on the hepatocytes, binds and removes LDL cholesterol (LDL-C) from circulation. The proprotein convertase subtilisin/kexin type 9 (PCSK9) inhibitor blocks the recycling of the LDLR in the liver.”
Author Response
Response to Reviewer 2
We thank the Reviewer for their time and thoughtful consideration in providing excellent feedback on important areas to improve the manuscript. We have addressed all of the points as described below, and we are grateful for your positive and extremely valuable evaluation.
MAJOR POINTS
To make the review more relevant to current guideline alterations, the authors should consider including information about some of the newer therapeutic regimen for diabetes and congestive heart failure such as SGLT2 inhibitors, as these are co-morbid conditions that also cause DKD.
Author Response: Thank you for the excellent suggestion!! SGLT-2 inhibitors are discussed in the section on ‘Glycemic Control.’ However, we have greatly expanded their discussion in general (under glycemic index) and with regards to other co-morbidities. Specifically, In the section about co-morbidities, in hypertension we have added text and also written a new section on heart failure as an additional co-morbidity when SGLT-2 inhibitors appear to be effective. We have also included a discussion of mineralocorticoid receptor antagonists and expanded on the discussion of GLP-1 agonists. Please see new text below:
2. Glycemic Index
The noninsulin options continue to expand in variety and each target a slightly different mechanisms in the glucose homeostasis pathway. Insulin sensitizers (biguanides) such as metformin, achieve glycemic control by inhibiting hepatic gluconeogenesis and opposing the actions of glucagon. Incretin agonists (GLP-1 receptor agonists) bind to receptors on pancreatic b-cells where they facilitate glucose-dependent insulin release. Renal sodium-glucose cotransporter-2 (SGLT-2) inhibitors lower blood sugar by preventing reabsorption of filtered glucose, resulting in excess sugar loss in the urine.
3.1. Hypertension
Other emerging therapies for hypertensive diabetics include SGLT-2 inhibitors, such as empagliflozin and dapagliflozin. As mentioned above, these agents are in use to help with diabetic glycemic control, however, there also appear to be a number of cardiovascular benefits. All current studies have shown significant reductions in systolic and diastolic blood pressure, with a similar efficacy in patients with compromised renal function. Furthermore, the EMPA-REG OUTCOME trial showed a demonstrated a 38% relative risk reduction in death from cardiovascular causes in patients with established cardiovascular disease who were administered empagliflozin vs. placebo.
3.3. Congestive Heart Failure
Guideline driven medical treatment of symptomatic heart failure includes pharmacological management with several agents that also display efficacy in achieving glycemic control and preventing CKD progression in type 2 diabetics. In 3 independent clinical trials, SGLT-2 inhibitors demonstrated a significant (20% to 25%) relative reduction in hospitalization for heart failure or cardiovascular death when compared with placebo. This benefit was consistent among patients with or without type 2 diabetes and in those experiencing CKD. Patients also reported a marked improvement in Health-related quality of life surveys.
Overactivation of the mineralocorticoid receptor is a common finding in both cardiovascular and renal disease. Finerenone is a selective nonsteroidal mineralocorticoid receptor antagonist that is now also used as a standard therapy for symptomatic congestive heart failure. Notably, there is increasing evidence that finerenone can prevent heart failure exacerbations and associated hospitalization in type 2 diabetic patients with various stages and severities of CKD. Furthermore, a reduction in other adverse cardiovascular outcomes including death from cardiovascular causes, myocardial infarction, or stroke was observed in patients taking finerenone.
3.4. Metabolic Syndrome
In addition to localization in pancreatic islets, incretin receptors have a wide range of tissue distribution, including the vasculature, cardiac myocytes, adipose tissue, gastrointestinal tract, hepatocytes, and central and peripheral nervous systems. Thus, in addition to augmentation of insulin secretion, GLP-1 agonists appear to have pleiotropic and beneficial effects on a wide range of metabolic functions. Their efficacy in metabolic syndrome, a common co-morbidity in type 2 diabetes may be attributed to many mechanisms. GLP-1 agonists are effective in altering blood lipid metabolism in a manner that lowers circulating LDL-C and reduces atherosclerosis and associated cardiovascular disease. Furthermore, GLP-1 agonists have potent anti-inflammatory effects, which may explain their efficacy in cardiovascular fitness, prevention of stroke, alleviation of chronic pain, and ability to reduce circulating levels of liver enzymes and liver steatosis.
Several possible models have been proposed about mechanisms by which these GLP-1 agonists can display anti-obesogenic effects. In adipose GLP-1 receptor activation increases beta oxidation and can aid in the conversion of white adipose tissue to brown adipose tissue, thus promoting energy consumption. GLP-1 agonists also are effective in appetite suppression, as they delay gastric emptying, and are thought to have central actions in the arcuate nucleus of the hypothalamus in the stimulation of satiety.
MINOR POINTS
Reviewer Query: On page 4, line 168 and elsewhere in the same paragraph, “angiotensin II receptor (ARB) inhibitors” should be changed to “angiotensin II receptor blockers (ARBs).” The term “ARB inhibitors” is a double negative and inaccurate.
Author Response: Thank you. Angiotensin II receptor blockers (ARBs) has been used to replace ARB-inhibitors in the text.
Reviewer Query: On page 4, line 177, the authors should define what “high-intensity pharmacological blood pressure therapy” is. Also, a brief discussion about the mechanism for how it contributes to DKD should be provided.
Author Response: On page 4, the paragraph has been entirely rewritten to define high-intensity pharmacological therapy (concomitant use of an ACEi and ARB) and to clarify the adverse effects on renal function. We’ve also created a separate paragraph to distinguish and clarify the effects of high-intensity blood pressure goals (vs. standard of <140/90) from those of high-intensity pharmacological treatment. Please see revisions below:
There is an ongoing debate regarding the intensity level for blood pressure management in patients with DKD. The therapeutic target for systolic and diastolic pressures in general practice is <140/90 mmHg, but the ideal systolic and diastolic pressure levels are still undetermined. The observation that there exists an increased risk of death or ESRD of 6.7% for every 10 mm Hg increase in baseline systolic blood pressure prompted the consideration of whether high-intensity pharmacological antihypertensive therapy (concomitant use of an ACEi and ARB) could be beneficial. However, while standard antihypertensive therapy (ACEi or ARB) achieved a decrease in proteinuria and other primary renal endpoints, high-intensity pharmacological blood pressure therapy was associated with increased rates of hyperkalemia and acute kidney injury (AKI). Therefore, the general consensus is to avoid combination therapy with an ACEi and an ARB, due to high rates of adverse side effects.
It is not also clear whether therapy targeted at achieving lower blood pressure goals than the standard goal of <140/90 mmHg is indeed beneficial. Data from the previously mentioned ACCORD trial found that 3% of diabetes patients with no evidence of CKD at baseline were diagnosed with CKD after three years of a systolic blood pressure treatment target of less than 120 mm Hg, compared to 1% of patients with standard antihypertension therapy goals of <140/90 mmHg. Yet, the use of antihypertension therapy has been shown to reduce cardiovascular events, such as stroke risk and myocardial infarction. In summary, the data suggests that identifying an optimal blood pressure must consider the patient’s personal cardiovascular status and CKD stage, in order to minimize any deleterious risks.
Reviewer Query: On page 4, line 182, “…the use of hypertension therapy has shown…” should read “…the use of antihypertensive therapy has been shown…” The term “antihypertensive therapy” should also replace “hypertension therapy” in the preceding sentence.
Author Response: Thank you. We have replaced “hypertension therapy” in both places to read “antihypertensive therapy” and have inserted the word “shown” in the indicated spot.
Reviewer Query: On page 5, lines 214-217, the following two sentences should be reversed for clarity and context and read: “The low-density lipoprotein (LDL) receptor (LDLR), located on the hepatocytes, binds and removes LDL cholesterol (LDL-C) from circulation. The proprotein convertase subtilisin/kexin type 9 (PCSK9) inhibitor blocks the recycling of the LDLR in the liver.”
Author Response: Agreed- thank you for catching this breach in clarity! The 2 sentences on pg. 5 have been reversed.

Reviewer 3 Report
The main errors in the manuscript are the lack of a clear definition of diabetic nephropathy, the omission of key studies in the field, and the failure to address the role of inflammation in the progression of the disease.
On page 5, the manuscript discusses the benefits of exercise in patient care plans, especially for DKD treatment. It mentions that exercise offers whole-body, multi-system benefits, including improved weight management, increased aerobic and cardiovascular capacity, decreased inflammation, and muscle atrophy. These benefits can help delay the progression of chronic diseases among diabetes patients.
Finerenone is a nonsteroidal mineralocorticoid receptor antagonist (MRA), the data on this medication in diabetic kidney disease have not been discussed.
More discussions on SGLT2 inhibitor and GLP-1 agonist are needed
The manuscript also cites various studies and articles related to DKD, its pathogenesis, diagnosis, and treatment. Some of these include studies on the role of advanced glycation end products in the pathogenesis of chronic kidney disease, the impact of intensive glucose control on kidney outcomes in patients with type 2 diabetes, and the effects of intensive glucose control on microvascular outcomes in patients with type 2 diabetes.
Dietary Recommendations:
Protein Intake: Non-dialysis patients should consume 0.8 mg/kg body weight/day of protein. The upper limit, 1.3 mg/kg body weight/day of protein, is discouraged due to increased rates of albuminuria and CVD-associated mortality [Page 6].
Sodium Intake: A low salt diet is recommended with less than 2,300mg/day salt intake. Practical suggestions include reducing salt use, opting for sodium-free food options, and consuming fewer highly processed and prepackaged foods [Page 6].
Fatty Acids: Replace foods rich in trans-fatty and saturated-fatty acids with omega-3 and omega-9 options, including seafood (like salmon and mackerel), nuts (such as walnuts, almonds, cashews), and oils (like olive, almond, and avocado) [Page 6].
Dietary Patterns: Instead of focusing on individual minerals and nutrients, adopt comprehensive dietary patterns like the Mediterranean diet and the DASH diet. These diets emphasize fresh vegetables, fruits, unrefined carbohydrates, and seeds, offering a more nutritional alternative to the Western diet [Page 6].
Exercise Recommendations:
5. Exercise: Exercise should be an integral part of patient care plans due to its whole-body, multi-system benefits. Exercise can lead to improved weight management, increased aerobic and cardiovascular capacity, decreased inflammation, and muscle atrophy [Page 5].
Specific Interventions:
6. Polyunsaturated Fatty Acid Consumption: There's evidence suggesting that high PUFA and low SFA intake can lead to the regression of nephropathy in diabetes patients [Page 11].
Physical Exercise: Exercise has shown benefits for patients with chronic renal failure. It can lead to increased muscle function and walking capacity, especially in elderly predialysis patients [Page 11].
Long-Term Intensive Lifestyle Intervention: Such interventions can promote the improvement of Stage III Diabetic Nephropathy [Page 11].
Avoiding the Western Diet: The traditional Western diet, filled with unhealthy sugars, sodium, trans-fats, and carbohydrates, is associated with increased rates of obesity and chronic diseases. Adopting healthier dietary patterns can be more beneficial [Page 6].
Active Patient Participation: Instead of focusing on strict dietary guidelines, encouraging patients to adopt overall healthier patterns of consumption can be more effective [Page 6].
overall well written
Author Response
Response to Reviewer 3
We thank the Reviewer for their time and thoughtful consideration in providing excellent feedback on important areas to improve the manuscript. We have addressed all of the points as described below, and we are grateful for your positive and extremely valuable evaluation.
Reviewer Query: Finerenone is a nonsteroidal mineralocorticoid receptor antagonist (MRA), the data on this medication in diabetic kidney disease have not been discussed.
Author Response: We thank the reviewer for the recommendation for discussing finerenone in the manuscript. We have added the following text in a new section on CHF:
3.3. Congestive Heart Failure
Overactivation of the mineralocorticoid receptor is a common finding in both cardiovascular and renal disease. Finerenone is a selective nonsteroidal mineralocorticoid receptor antagonist that is now also used as a standard therapy for symptomatic congestive heart failure. Notably, several recent clinical trials have demonstrated that finerenone can prevent heart failure exacerbations and associated hospitalization in type 2 diabetic patients with various stages and severities of CKD. Furthermore, a reduction in other adverse cardiovascular outcomes including death from cardiovascular causes, myocardial infarction, or stroke was observed in patients taking finerenone.
Reviewer Query: More discussions on SGLT2 inhibitor and GLP-1 agonist are needed
Author Response: Thank you for the important suggestion!! We absolutely agree, and have therefore added the following sections to expand on SGLT-2 inhibitors and GLP-1 agonists more appropriately:
2. Glycemic Index
The noninsulin options continue to expand in variety and each target a slightly different mechanisms in the glucose homeostasis pathway. Insulin sensitizers (biguanides) such as metformin, achieve glycemic control by inhibiting hepatic gluconeogenesis and opposing the actions of glucagon. Incretin agonists (GLP-1 receptor agonists) bind to receptors on pancreatic b-cells where they facilitate glucose-dependent insulin release. Renal sodium-glucose cotransporter-2 (SGLT-2) inhibitors lower blood sugar by preventing reabsorption of filtered glucose, resulting in excess sugar loss in the urine.
3.2. Hypertension
Other emerging therapies for hypertensive diabetics include SGLT-2 inhibitors, such as empagliflozin and dapagliflozin. As mentioned above, these agents are in use to help with diabetic glycemic control, however, there also appear to be a number of cardiovascular benefits. All current studies have shown significant reductions in systolic and diastolic blood pressure, with a similar efficacy in patients with compromised renal function. Furthermore, the EMPA-REG OUTCOME trial showed a demonstrated a 38% relative risk reduction in death from cardiovascular causes in patients with established cardiovascular disease who were administered empagliflozin vs. placebo.
3.3. Congestive Heart Failure
Guideline driven medical treatment of symptomatic heart failure includes pharmacological management with several agents that also display efficacy in achieving glycemic control and preventing CKD progression in type 2 diabetics. In 3 independent clinical trials, patients taking SGLT-2 inhibitors demonstrated a significant (20% to 25%) relative reduction in hospitalization for heart failure or cardiovascular death when compared with the placebo group. This benefit was consistent among patients with or without type 2 diabetes and in those experiencing CKD. Patients also reported a marked improvement in Health-related quality of life surveys.
3.4. Metabolic Syndrome
In addition to localization in pancreatic islets, incretin receptors have a wide range of tissue distribution, including the vasculature, cardiac myocytes, adipose tissue, gastrointestinal tract, hepatocytes, and central and peripheral nervous systems. Thus, in addition to augmentation of insulin secretion, GLP-1 agonists appear to have pleiotropic and beneficial effects on a wide range of metabolic functions. Their efficacy in metabolic syndrome, a common co-morbidity in type 2 diabetes may be attributed to many mechanisms. GLP-1 agonists are effective in altering blood lipid metabolism in a manner that lowers circulating LDL-C and reduces atherosclerosis and associated cardiovascular disease. Furthermore, GLP-1 agonists have potent anti-inflammatory effects, which may explain their efficacy in cardiovascular fitness, prevention of stroke, alleviation of chronic pain, and ability to reduce circulating levels of liver enzymes and liver steatosis.
Several possible models have been proposed about mechanisms by which these GLP-1 agonists can display anti-obesogenic effects. In adipose GLP-1 receptor activation increases beta oxidation and can aid in the conversion of white adipose tissue to brown adipose tissue, thus promoting energy consumption. GLP-1 agonists also are effective in appetite suppression, as they delay gastric emptying, and are thought to have central actions in the arcuate nucleus of the hypothalamus in the stimulation of satiety.

Round 2
Reviewer 2 Report
Authors did a sufficient job in addressing all reviewer comments in the revised manuscript.
Reviewer 3 Report
It appears that all comments have been appropriately responded to. I have no further comments and recommend publication.